# Spatiotemporal dynamics and epidemiological impact of SARS-CoV-2 XBB lineage dissemination in Brazil in 2023

Ighor Arantes,[1] Marcelo Gomes,[2] Kimihito Ito,[3] Sharbilla Sarafim,[1] Tiago Gräf,[4] Fabio Miyajima,[5] Ricardo Khouri,[6] Felipe Cotrim de Carvalho,[7] Walquiria Aparecida Ferreira de Almeida,[7] Marilda Mendonça Siqueira,[8] Paola Cristina Resende,[8] Felipe Gomes Naveca,[1,9] Gonzalo Bello,[1] COVID-19 Fiocruz Genomic Surveillance Network

**ABSTRACT** The SARS-CoV-2 XBB is a group of highly immune-evasive lineages of the Omicron variant of concern that emerged by recombining BA.2-descendent lineages and spread worldwide during 2023. In this study, we combine SARS-CoV-2 genomic data ($n$ = 11,065 sequences) with epidemiological data of severe acute respiratory infection (SARI) cases collected in Brazil between October 2022 and July 2023 to reconstruct the space-time dynamics and epidemiologic impact of XBB dissemination in the country. Our analyses revealed that the introduction and local emergence of lineages carrying convergent mutations within the Spike protein, especially F486P, F456L, and L455F, propelled the spread of XBB* lineages in Brazil. The average relative instantaneous reproduction numbers of XBB* + F486P, XBB* + F486P + F456L, and XBB* + F486P + F456L + L455F lineages in Brazil were estimated to be 1.24, 1.33, and 1.48 higher than that of other co-circulating lineages (mainly BQ.1*/BE*), respectively. Despite such a growth advantage, the dissemination of these XBB* lineages had a reduced impact on Brazil's epidemiological scenario concerning previous Omicron subvariants. The peak number of SARI cases from SARS-CoV-2 during the XBB wave was approximately 90%, 80%, and 70% lower than that observed during the previous BA.1*, BA.5*, and BQ.1* waves, respectively. These findings revealed the emergence of multiple XBB lineages with progressively increasing growth advantage, yet with relatively limited epidemiological impact in Brazil throughout 2023. The XBB* + F486P + F456L + L455F lineages stand out for their heightened transmissibility, warranting close monitoring in the months ahead.

**IMPORTANCE** Brazil was one the most affected countries by the SARS-CoV-2 pandemic, with more than 700,000 deaths by mid-2023. This study reconstructs the dissemination of the virus in the country in the first half of 2023, a period characterized by the dissemination of descendants of XBB.1, a recombinant of Omicron BA.2 lineages evolved in late 2022. The analysis supports that XBB dissemination was marked by the continuous emergence of indigenous lineages bearing similar mutations in key sites of their Spike protein, a process followed by continuous increments in transmissibility, and without repercussions in the incidence of severe cases. Thus, the results suggest that the epidemiological impact of the spread of a SARS-CoV-2 variant is influenced by an intricate interplay of factors that extend beyond the virus's transmissibility alone. The study also underlines the need for SARS-CoV-2 genomic surveillance that allows the monitoring of its ever-shifting composition.

**KEYWORDS** SARS-CoV-2, Brazil, XBB, phylogeography, SARI

S ince its emergence in late 2021, the SARS-CoV-2 variant of concern (VOC) Omicron has not only replaced previous VOCs but has also undergone rapid evolutionary

Address correspondence to Ighor Arantes, ighorarantes@gmail.com, or Gonzalo Bello, gbello@ioc.fiocruz.br.

The authors declare no conflict of interest.

See the funding table on p. 14.

*[This article was published on 5 February 2024 with missing information in Acknowledgments. The Acknowledgments were corrected in the current version, posted on 5 March 2024.]*

changes, giving rise to a multitude of sub-lineages that spread around the world (1). The XBB lineage is one of these Omicron sub-lineages that emerged around July 2022 by the recombination of BA.2-descendent lineages BJ.1 and BM.1.1.1. Its initial dissemination primarily occurred in Asia, with Singapore witnessing a sharp upsurge in cases attributed to this strain (2–5). Although the XBB lineage demonstrated remarkable immune evasiveness and an effective reproduction number ($R_e$) ~1.2 times higher than those of parental variants (5–20), it displayed a weaker angiotensin-converting enzyme-2 (ACE2) binding affinity than other immune evasive Omicron subvariants that circulated during the latter half of 2022, such as the BQ.1* lineages prevalent in Western countries (5, 21, 22) .

In late 2022, a new XBB lineage designated XBB.1.5 arose and rapidly spread worldwide. The XBB.1.5 lineage harbors one additional mutation (S:F486P) in the receptor-binding domain (RBD) of the Spike that significantly improves the XBB's ability to attach to the human ACE2 receptor without eroding the variant's ability to evade humoral immunity (21, 23–27). The $R_e$ of XBB.1.5 was estimated to be ~1.2 times greater than that of the parental lineage XBB.1 and outcompeted BQ.1* lineages (24). Throughout 2023, multiple XBB lineages carrying mutation S:F486P evolved independently in different countries (1). This convergent evolution is the case of the lineage XBB.1.16 that initially spread in India and displayed similar characteristics to lineage XBB.1.5 regarding host cell entry efficiency, neutralization evasion, and growth advantage (17, 28, 29). Thus, increased immune evasion and high receptor-binding affinity resulted in a substantial growth advantage of XBB* + S:F486P relative to previously circulating Omicron lineages.

In 2023, multiple XBB* + F486P subvariants acquired an additional convergent mutation S:F456L in the RBD of the Spike and displayed a growth advantage relative to the XBB variant precursors (30–32). Mutation S:F456L reduces the ACE2 binding affinity but increases the variant's immune evasion capabilities, even after previous XBB infection (17, 31, 33–36). The most prevalent lineage with this mutation is EG.5, which evolved from the lineage XBB.1.9.2 (S:F486P), showing one of the fastest growth rates among XBB lineages circulating in North America and Europe in early 2023 (30, 37–39). By late March 2023, another group of XBB* + F486P lineages harboring mutations S:L455F and S:F456L, such as lineages XBB.1.5.70, GK*, JD.1*, and HK.3, were detected for the first time and are rapidly spreading (40). These lineages exhibit higher immune evasion than XBB* + F486P lineages and heightened ACE2 binding affinity relative to XBB* + F486P + F456L lineages (26, 41), which may thus lead to further growth advantage.

Brazil was one of the most heavily affected countries in the world by the SARS-CoV-2 pandemic. However, little is known about the molecular evolution trajectory of XBB lineages circulating in the country. In this study, we analyzed the temporal transition and transmission advantage of XBB lineages that spread in Brazil during the first half of 2023. We reconstruct the spatiotemporal spread dynamics of major XBB* + F486P, XBB* + F486P + F456L, and XBB* + F486P + F456L + L455F lineages circulating in Brazil to assess the epidemiological impact of their dissemination by analyzing the hospitalizations related to severe acute respiratory illness (SARI) due COVID-19, over the Omicron waves.

## MATERIALS AND METHOD

### SARS-CoV-2 Brazilian genome sequences

A total of 11,065 SARS-CoV-2 complete genome sequences recovered across all 26 Brazilian states plus the country's Federal District between 1 October 2022 and 31 July 2023 were newly generated by the COVID-19 Fiocruz Genomic Surveillance Network (42). All samples had real-time RT-PCR cycling threshold below 30, indicating elevated viral load. There was no intended bias considering dense sampling of specific outbreaks, and we do not consider the health status (severe, mild, or asymptomatic) or epidemiological data (gender and age) of individuals as a criterion for sampling. SARS-CoV-2 genome sequences were generated using the Illumina COVIDSeq Test kit as previously described (43). Raw data were converted to FASTQ files at Illumina BaseSpace cloud, and consensus sequences were produced with the most up-to-date version of DRAGEN COVID LINEAGE

or ViralFlow 1.0 (44). All genomes were evaluated for mutation calling and quality with the Nextclade 2.14.0 algorithm (45) and were uploaded to the EpiCoV database of GISAID (https://gisaid.org/) (see the supplemental material). Additionally, we downloaded all publicly available Brazilian sequences (*n* = 14,822) collected since 1 October 2022 that were submitted to the EpiCov database until 31 July 2023 with complete collection date and lineage assignment. Whole-genome consensus sequences were classified using the "Phylogenetic Assignment of Named Global Outbreak Lineages" (PANGOLIN) software v4.3 (pangolin-data version 1.21) (46).

## Non-Brazilian XBB genome sequences and data set composition

To elucidate the spatiotemporal dynamics of prominent Brazilian XBB lineages that exhibited the S:F486P, S:F456L, and S:L455F mutations, we retrieved all non-Brazilian sequences pertaining to lineages XBB.1.4.2 (*n* = 16), XBB.1.5.86 + HA.1 (*n* = 478), XBB.1.18* + FE.1* (*n* = 1,772), XBB.1.5.70 + GK* (*n* = 350), and XBB.1.5.102 + JD.1* (*n* = 174) that were submitted to the EpiCov database until 31 July 2023 (see the supplementarl material). To obtain a final data set for each studied lineage, all genomes, whether Brazilian or foreign, were filtered based on (i) the presence of all described synapomorphic mutations of their respective lineages, (ii) a complete collection date, and (iii) a small fraction of unidentified positions ($N < 5\%$). To enhance the accuracy of the root position in our phylogenetic reconstructions, we obtained all high coverage ($N < 1\%$) genomes with complete collection dates from ancestral lineages for potential Brazilian ones, namely, XBB.1 (*n* = 811), XBB.1.5 (*n* = 3,886), and XBB.1.4 (*n* = 12). To streamline our computational analyses without compromising lineage diversity, we clustered them, when needed, with CD-HIT (47), leading to data sets of much-reduced dimensions (XBB.1, *n* = 3, and XBB.1.5, *n* = 118).

## Maximum-likelihood (ML) phylogenetic analysis

Seven XBB data sets of 29,421 nt (from position 1 of ORF1ab to position 117 of ORF10) were produced by a combination of Brazilian and non-Brazilian sequences belonging to the following lineages: (i) XBB.1.18* + FE.1 (*n* = 1,843), (ii) FE.1.1 (*n* = 1,251), (iii) FE.1.2 (*n* = 1,051), (iv) XBB.1.5.102 (*n* = 1,522), (v) XBB.1.4.2 (*n* = 132), (vi) XBB.1.5.86 + HA.1 (*n* = 1,010), and (vii) XBB.1.5.70 + GK* + JD.1 (*n* = 716). These were aligned using MAFFT v7.467 (48) and manually curated when needed. The aligned data sets were subjected to ML phylogenetic analysis using IQ-TREE v2.2.2.7 (49) under the general time-reversible model of nucleotide substitution with a gamma-distributed rate variation among sites, four rate categories (G4), a proportion of invariable sites (I), and empirical base frequencies (F) nucleotide substitution model, as selected by the ModelFinder application embedded in IQ-TREE (50). The approximate likelihood-ratio test (aLRT) assessed the branch support based on the Shimodaira–Hasegawa-like procedure with 1,000 replicates.

## Bayesian time-scaled phylogenetic and phylogeographic analyses

The temporal signal of the XBB data sets was assessed from the ML trees by performing a regression analysis of the root-to-tip divergence against sampling time using TempEst v1.5.36 (51). Sequences that diverged more than 1.5 interquartile ranges from the root-to-tip regression were considered outliers and removed from the analysis. The significance of the association between the two variables was assessed with a Spearman correlation test implemented in the R programming language v.4.1.2. Time-scaled Bayesian phylogenetic trees were estimated in BEAST v1.10.4 (52) under a strict molecular clock model with a uniform distribution between 5.0E-4 and 1.0E-3 and a non-parametric Bayesian skyline model as the coalescent tree prior (53). Due to the large size of some of our data sets (*n* > 500 sequences), time-scaled trees were reconstructed using a modified version of BEAST (https://beast.community/thorney_beast) that incorporates the possibility of a fixed ML topology. This method alleviates the

computational burden by rescaling the input tree branch lengths based on (i) the clock and (ii) coalescent models. The ML phylogenetic trees were inputted in the BEAST XML file as a starting point, and data trees and analyses were performed as specified above. For phylogeographic analyses, a set of 1,000 trees was randomly selected from the posterior distribution of trees resulting from the temporal BEAST analysis. Sampling locations were used as traits in the phylogeographic model, and the ancestral states were reconstructed using a discrete symmetric model (54). Samples were grouped by continental regions, except Brazilian sequences that were grouped in a separate location. Major Brazilian transmission clusters were defined as highly supported (aLRT > 0.75) monophyletic clades that descend from an most recent common ancestor (MRCA) node probably located [posterior state probability (PSP) >0.75] in Brazil. Markov chain Monte Carlo (MCMC) simulations were run sufficiently long to ensure convergence [effective sample size (ESS) >200] in all parameter estimates as assessed in TRACER v1.770 (55). The maximum clade credibility trees were summarized with TreeAnnotator v1.10 and visualized using treeio v3.1.7 (56) and ggtree v3.2.1 (57) .

## Relative instantaneous reproduction number ($R_{RI}$) estimations

We measured the transmission advantage of XBB lineages carrying the S:F486P, S:F486P + F456L, and S:F486P + F456L + L455F mutations compared to other lineages using their $R_{RI}$. The details of the estimation method have been described elsewhere (58). Briefly, the $R_{RI}$ is an effective reproduction number representing the average number of people an infected individual at time $t$ could be expected to infect, given that conditions remain unchanged (59). Suppose that lineage $A_1, ..., A_n$ , and $a$ are circulating in the population. The lineage $a$ is used as the baseline of the comparison. Let $R_{A_1}(t), ..., R_{A_n}(t)$ , and $R_a(t)$ be the $R_{RI}$ of lineages $A_1, ..., A_n$ , and $a$ at time $t$, respectively. Assuming the rate of $R_{A_i}(t)$ to $R_a(t)$ is constant over time for each lineage $A_i (1 \leq i \leq n)$ , the effective reproduction number of $A_i$ can be written as follows:

$$R_{A_i}(t) = k_i R_a(t),$$

where $k_i$ is a constant. This constant $k_i$ is called the $R_{RI}$ of lineage $A_i$ with respect to (w.r.t.) lineage $a$. A previous study (58) has shown that observation data on the SARS-COV-2 variant replacement were well-modeled by assuming that the rate of the effective reproduction number of a variant to that of another variant was constant over time. It was also shown that the time evolution of (relative) frequencies of lineages $A_1, ..., A_n$ , and $a$ among a viral population is well-modeled without the number of infections (60, 61). Let $q_{A_i}(t)$ and $q_a(t)$ be the frequencies of lineages $A_i$ and $a$, respectively. Then, $q_{A_i}(t)$ and $q_a(t)$ are represented by the following formula:

$$q_{A_i}(t) \quad \frac{k_i \sum_{j=1}^{l} g(j) q_{A_i}(t-j)}{\sum_{j=1}^{l} g(j) q_a(t-j) + \sum_{i}^{n} k_i \sum_{j=1}^{l} g(j) q_{A_i}(t-j)},$$

$$q_a(t) \quad 1 - \sum_{i=1}^{n} q_{A_i}(t),$$

where $g(j)$ is the probability mass function of generation time distribution and $l$ is the day when the probability of generating new infections becomes negligibly small. Park et al. estimated the Omicron strains' mean generation time to be 3.0 using within-household transmission data (62). The generation time distribution they estimated was a lognormal distribution with a log mean of 0.98 and a log standard deviation of 0.47, of which the mean is 2.97, and the variance is 2.19. Using these values, the probability mass function of the generation time $g(j)$ was modeled by discretizing the gamma distribution having the same mean and variance as follows:

$$g(j) = \begin{cases} 0, & \text{if } j = 0 \\ \frac{\text{CDF}(\text{Gamma}(\alpha, \theta), j) - \text{CDF}(\text{Gamma}(\alpha, \theta), j-1)}{\text{CDF}(\text{Gamma}(\alpha, \theta), l)}, & \text{if } 1 \leq j \leq l \\ 0, & \text{if } j > l \end{cases}$$

where $\alpha$ and $\theta$ are the shape and scale parameters of gamma distribution (Gamma) and the CDF is the cumulative distribution function up to the given point. Parameters of the gamma distribution were set to be $\alpha = 4.03$ and $\theta = 0.737$, so that the gamma distribution has the same mean and variance to the lognormal distribution estimated by Park et al. (62).

We consider XBB* lineages carrying S:F486P, S:F486P + F456L, and S:F486P + F456L + L455F mutations as $A_1$, $A_2$, and $A_3$, respectively, and other lineages (mainly BQ.1*/BE*) were used as the baseline lineage $a$. We assume that viruses of these lineage groups in a region were sampled according to a Dirichlet multinomial distribution with the shape parameters $q_{A_1}(t)M$, $q_{A_2}(t)M$, $q_{A_3}(t)M$, and $q_a(t)M$ at time $t$ where $M$ represent the sum of the Dirichlet distribution parameters. We also assume that the relative reproduction numbers of XBB* + S:F486P, XBB* + S:F486P + F456L, and XBB* + S:F486P + F456L + L455F w.r.t. others are the same in all regions. Lineage groups observed in each region were counted using weekly bins starting from Monday. By maximizing the likelihood function of the multinomial distribution in all regions, we estimated the relative reproduction numbers $k_1$, $k_2$, and $k_3$; the sum of parameters Dirichlet distribution parameters $M$; and hypothetical initial frequencies $q_{A_1}(t_0)$, $q_{A_2}(t_0)$, $q_{A_3}(t_0)$ in each region on 3 October 2022. The estimation of the date when each lineage group was introduced in each region is not the purpose of this specific analysis. Instead, we estimated hypothetical initial frequencies of each one, $q_{A_i}(t_0)$, if it existed in each region on 3 October 2022. Thus, a late introduction of a lineage $A_i$ to a region results in a small estimate of $q_{A_i}(t_0)$ in the region. The 95% confidence intervals (CI) of estimates were estimated by profile likelihood. The trajectories of variant frequencies were calculated using the maximum-likelihood estimates of the parameters in the model. The 95% CI of the trajectories was calculated using combinations of parameters within the 95% confidence region (63).

## Data on hospitalizations for SARI

We extracted data about hospitalizations resulting from SARI attributed explicitly to SARS-CoV-2 (SARI-COVID) in Brazil during the period spanning from January 2022 to July 2023. This information was sourced from the Influenza Surveillance Information System (SIVEP-Gripe) database (https://opendatasus.saude.gov.br/dataset?tags=SRAG). To identify SARI cases, we utilized a set of four criteria which required individuals to exhibit: (i) fever, including self-reported cases; (ii) cough or sore throat; (iii) dyspnea or oxygen saturation levels below 95% or experiencing respiratory discomfort; and (iv) hospitalization. Once an individual meets these criteria and is admitted to a hospital for SARI, their case must be reported and recorded as a distinct entry in the SIVEP-Gripe database. Confirmation of hospitalization for SARI-COVID was contingent on a positive result from the RT-PCR molecular test for SARS-CoV-2. For our analysis, we considered all the records within the SIVEP-Gripe database that adhered to the criteria for defining a hospitalized SARI case. We excluded records related to non-hospitalized deaths from our examination.

## RESULTS

### Circulation of SARS-CoV-2 Omicron subvariants in Brazil in 2023

We analyzed 25,887 Brazilian SARS-CoV-2 genomes collected across all states between 1 October 2022 and 31 July 2023 (Fig. 1A). Sequences were mostly classified as XBB* (28%) or BA.5* (72%) lineages. While most BA.5* sequences were from the Southeast region (Fig. 1B), most XBB* sequences were collected in the North region (Fig. 1C). The

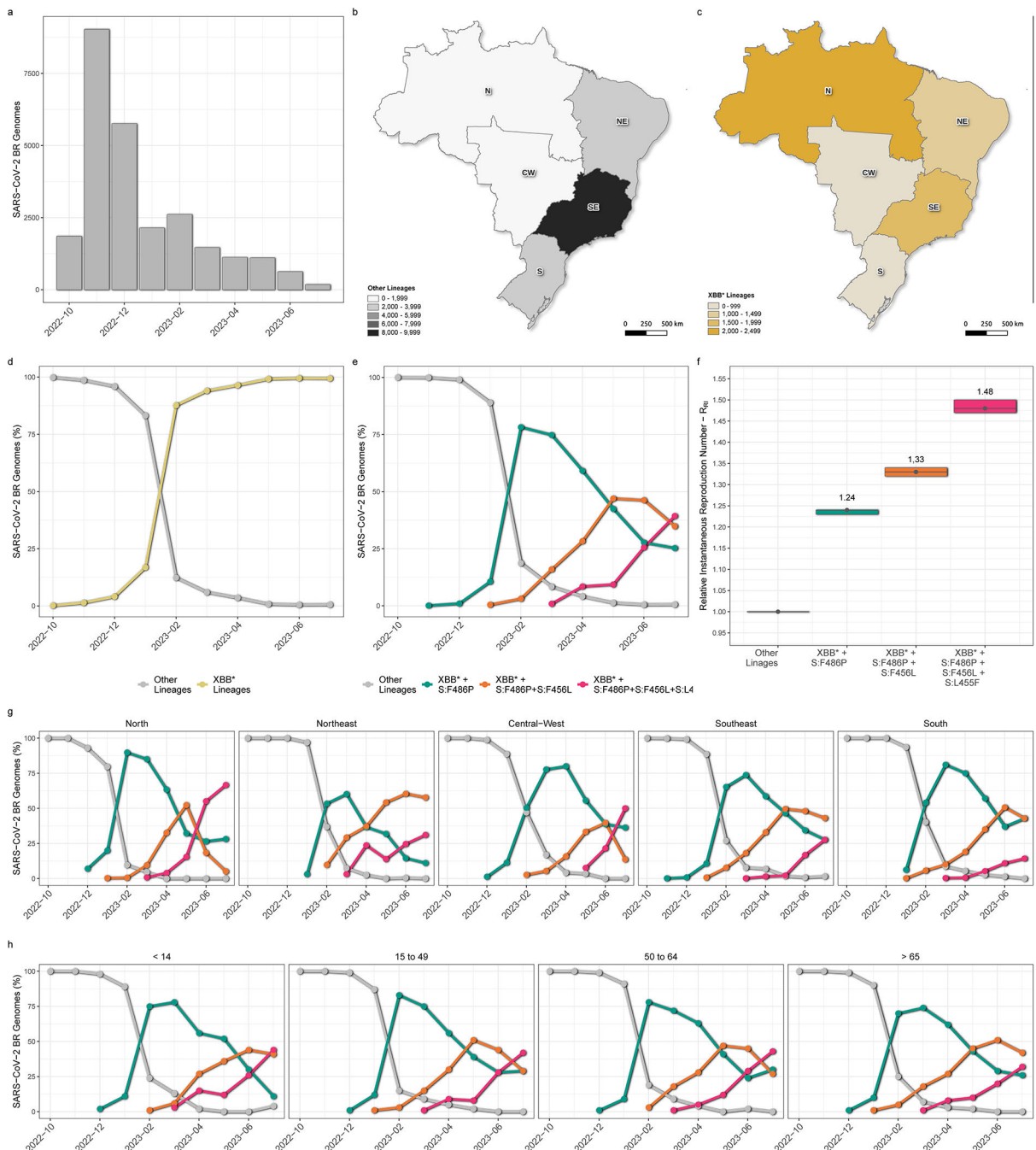

**FIG 1** Space-time evolution of XBB subvariants in Brazil between October 2022 and July 2023. (A) Monthly distribution of the 25,887 Brazilian SARS-CoV-2 genomes that constituted the data set for this study. (B and C) Spatial distribution of SARS-CoV-2 genomes across the five Brazilian regions of the 18,518 non-XBB (B) and the 7,369 XBB genomes (C). (D) Temporal fluctuations in the prevalence of both XBB and non-XBB variants in Brazil. (E) Evolution of distinct XBB subvariants groups relative prevalence in Brazil. The groups are defined by the mutations S:F486P, S:F456L, and S:L455F, all located in the RBD region of the Spike protein. Genomes of XBB subvariants in which all three mutations were absent and those of non-XBB lineages were grouped in the "others" category. (F) Relative instantaneous reproductive number ($R_{RI}$) of XBB* groups w.r.t. the "others" category. (G and H) Evolution of distinct XBB subvariants groups relative prevalence across the five Brazilian regions (G) and distinct age groups (H). Maps were generated with QGIS v.3.10.2 software (http://qgis.org) using public access data downloaded from the GADM v.3.6 database (https://gadm.org) and shapefiles obtained from the Brazilian Institute of Geography and Statistics (https://portaldemapas.ibge.gov.br/portal.php#homepage).

XBB* sequences were first detected in Brazil in October 2022 and circulated at a low prevalence (<10%) during that year. However, their proportion experienced a rapid surge

in 2023 going from ~15% to ~90% between January and February (Fig. 1D). The XBB* sequences were categorized in four groups based on mutations at the RBD region of the Spike protein: XBB*, XBB* + F486P, XBB* + F486P + F456L, and XBB* + F486P + F456L + L455F, which were indeed the only Spike mutations with pronounced increase (>25%) in prevalence over time across all XBB* genomes during the study period (Fig. S1). The temporal analysis revealed a clear pattern of substitution of XBB groups over time (Fig. 1E). The XBB* group was first detected in October 2022 and displayed a very low prevalence (~5%) over the studied period. The dominance of XBB lineages in the Brazilian epidemic in 2023 coincides with the introduction and dissemination of the XBB* + F486P group that was first detected by November 2022 and reached its peak in relative prevalence by February 2023. The XBB* + F486P + F456L and XBB* + F486P + F456L + L455F groups emerged later but progressively assumed dominance in the country. The XBB*+F486P + F456L group was first detected by January 2023 and reached its zenith in relative prevalence by May 2023, whereas the XBB* + F486P + F456L + L455F group was first detected by March 2023 and surpassed the XBB* + F486P and XBB* + F486P + F456L groups 3 and 4 months later, respectively. The relative instantaneous reproduction number ($R_{RI}$) of XBB* lineages carrying additional Spike mutations w.r.t. "other" lineages circulating in Brazil in 2023 was calculated using the renewal equation-based model developed by Ito et al. (60) (Fig. 1F). The average $R_{RI}$ of XBB* + F486P, XBB* + F486P + F456L and XBB* + F486P + F456L + L455F lineages in Brazil were estimated to be 1.24, 1.33 and 1.48 higher than that of other co-circulating lineages (mainly BQ.1*/BE*), respectively. Thus, the average $R_{RI}$ of XBB* + F486P + F456L lineages was 1.07 times higher than that of XBB* + F486P lineages, and the average $R_{RI}$ of XBB* + F486P + F456L + L455F lineages was 1.19 and 1.11 times higher than that of XBB* + F486P and XBB* + F486P + F456L lineages, respectively. The same pattern of XBB subvariants substitutions over time was consistently observed in the whole country, although the current prevalence of the different XBB groups may vary across Brazilian regions (Fig. 1G) and across all age groups (Fig. 1H).

## Emergence and dissemination of major XXB lineages in Brazil

The assignment of dynamic Pango lineages to the Brazilian XBB genomes revealed a heterogeneous composition within each of the groups and further revealed that mutations S:F486P/F456L/L455F arose independently in multiple lineages (Fig. 2A). The XBB* + F486P group was mainly composed of lineages XBB.1.5.102 (34%), XBB.1.18.1 (19%), XBB.1.5.86 (12%), XBB.2.3* (3%), XBB.1.4.2 (2%), and others XBB.1.5-descendant lineages (~25%) (Fig. 2B). The XBB* + F486P + F456L group was mostly composed of XBB.1.18.1-descendant lineages such as FE.1.2 (~50%), FE.1.1 (~20%), and FE.1 (~10%) (Fig. 2B). The XBB* + F486P + F456L + L455F group was mostly composed of lineage XBB.1.5.70 (~50%) and their descendent lineages GK.1 (~30%) and GK.1.1 (~5%) and by the XBB.1.5.102-descendent lineage JD.1 (~5%) and JD.1.2 (~10%) (Fig. 2B). The relative frequency of the different XBB lineages within each group varied across Brazilian regions, particularly in the Northern region (Fig. 2C). There, the most prevalent lineage in the first group was the XBB.1.5.102 (~80%), in the second group was an XBB.1.18.1 + F456L lineage distinct of FE.1* (~40%), while in group 3 prevails the JD.1* (~40%). All these three variants were scarcely detected in other regions of the country.

Until the end of August 2023, among SARS-CoV-2 genomes submitted to GISAID (https://gisaid.org/), some of the major XBB* lineages detected in Brazil remained mostly restricted to its borders, while others were frequently detected outside of the country. Within the XBB* + F486P group, most genomes (55-85%) of lineages XBB.1.18.1, XBB.1.5.86, XBB.1.5.102, and XBB.1.4.2 were sampled in Brazil, while most genomes (>95%) of lineages XBB.1.5 and XBB.2.3* were sampled outside the country. Within the XBB* + F486P + F456L group, lineages FE.1 and FE.1.2 were equivalently sampled in Brazil and abroad (~50%), while lineages FE.1.1 and HA.1 were mainly detected outside Brazil (75-80%). Within the XBB* + F486P + F456L + L455F group, JD.1 was mostly sampled in Brazil (~85%), XBB.1.5.70 and JD.1.2 were equivalently sampled in Brazil and abroad

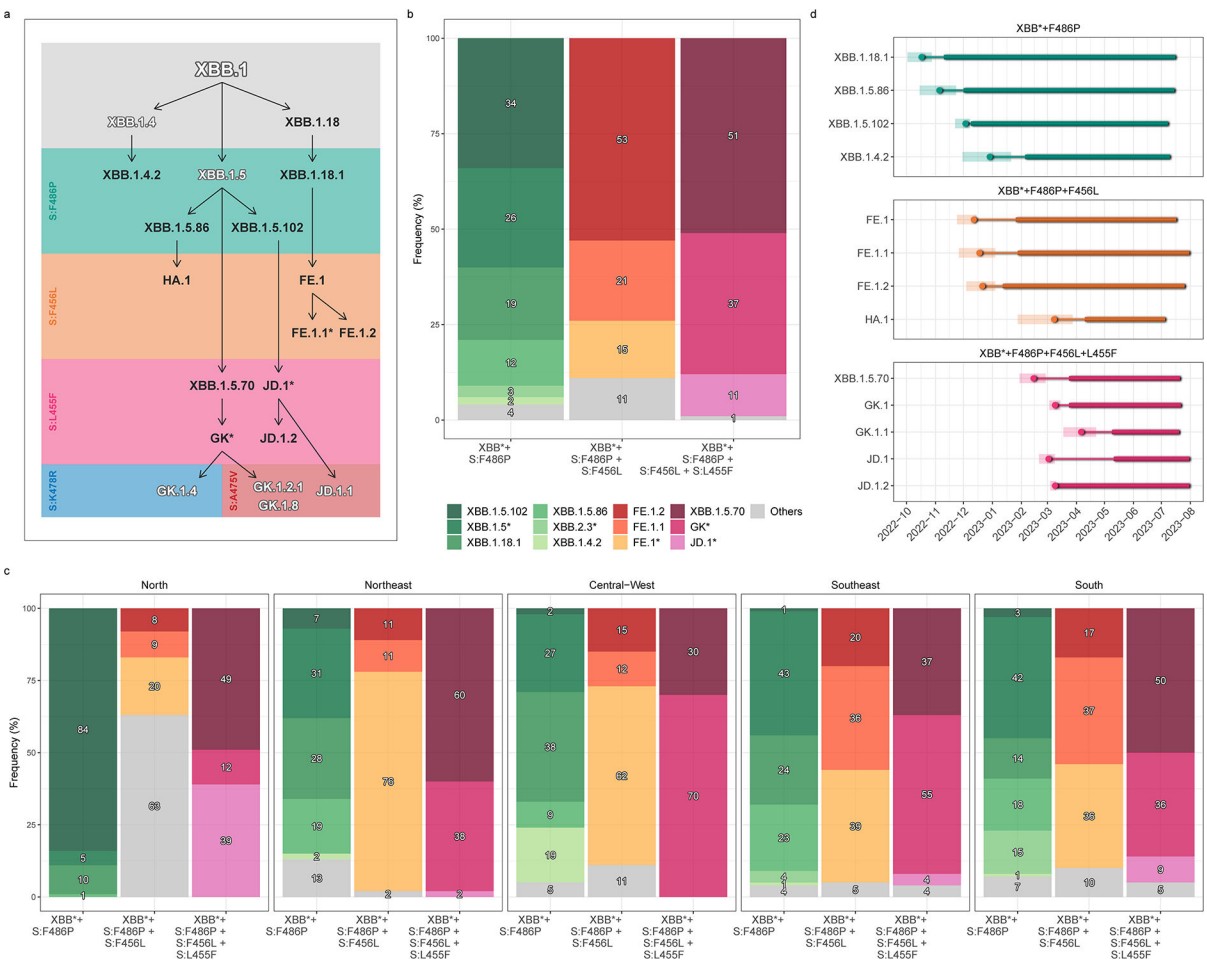

**FIG 2** Major XBB* lineages that arose and spread in Brazil. (A) Fluxogram representing the major XBB* Pango lineages that emerged and spread in Brazil, focused on the sequential accumulation of key Spike protein mutations (S:F486P, S:F456L, and S:L455F). Lineages that evolved in Brazil are in dark gray, while foreign lineages or those not included in out phylogeographic analysis are in white with a gray halo. (B and C) Relative frequency of major Pango XBB* lineages bearing the S:F486P, S:F486P + F456L, and S:F486P + F456L + L455F mutations in Brazil (B) and in different country regions (C). (D) Temporal dynamics of major Pango XBB* lineages that emerged in Brazil. For each XBB* lineage, we represent the time of the most recent common ancestor (TMRCA, circle) and its 95% highest posterior density (HPD) interval (transparent polygon), the period of cryptic circulation (thinner line), and the sampling range (thicker line). All represented Pango XBB* lineages were inferred as having their MRCA most probably located in Brazil (PSP > 0.75).

(~50%), and lineages GK.1 and GK.1.1 were mostly sampled outside the country (60%–90%).

In order to ascertain the most probable geographic origin of major XBB* lineages detected in Brazil, we conducted Bayesian phylogeographic inferences by compiling all SARS-CoV-2 complete genomes pertaining to each aforementioned XBB lineage that were available in EpiFlu GISAID by the end of July 2023. Our phylogeographic analyses pinpointed Brazil as the likely origin (PSP > 75%) of lineages XBB.1.4.2, XBB.1.5.70, XBB.1.5.86, XBB.1.5.102, XBB.1.18, XBB.1.18.1, FE.1, FE.1.1, FE.1.2, GK.1, GK.1.1, JD.1, and JD.1.2 (Fig. S2 through S7; Table S1). Brazilian autochthonous lineages carrying the S:F486P mutation arose between mid-October (XBB.1.18.1) and late December (XBB.1.4.2) of 2022, followed by Brazilian lineages harboring S:F486P + F456L mutations from mid-December 2022 (FE.1) to early March 2023 (HA.1) and Brazilian lineages harboring S:F486P + F456L + L455F mutations from mid-February (XBB.1.5.70) to early April (GK.1.1) of 2023 (Fig. 2D; Table S1). Some of these XBB lineages spread beyond Brazil's borders and gave rise to new sub-lineages in other locations, including multiple ones in North America, such as GK.1.2 (PSP = 0.90), GK.1.3 (PSP = 1.0), GK.2 (PSP = 0.70), GK.3 (PSP = 1.0), and JD.1.1 (PSP = 1.0) (Fig. S7).

## Epidemiological impact of dissemination of XXB in Brazil

To assess the epidemiological consequences of the proliferation of XBB lineages within the context of Omicron waves in Brazil, we analyzed the temporal trends of hospitalizations attributed to SARI-COVID from January 2022 to July 2023. The available data reveal a recurring pattern throughout 2022 characterized by three discernible waves occurring approximately 6 months apart, reaching their peaks in January, July, and December, under the dominance of BA.1*, BA.5*, and BQ* subvariants, respectively (Fig. 3A). These waves displayed a trend of diminishing magnitude over time, experiencing reductions of approximately 75% and 35%, respectively, compared to their preceding counterparts. As we enter the phase where XBB constitutes the most prevalent subvariant in the country in early 2023, there was a discrete increase in SARI-COVID cases around early March coinciding with the dissemination of the XBB* + F486P group, followed by a period of decline that persists throughout the dissemination of the XBB* + F486P + F456L and

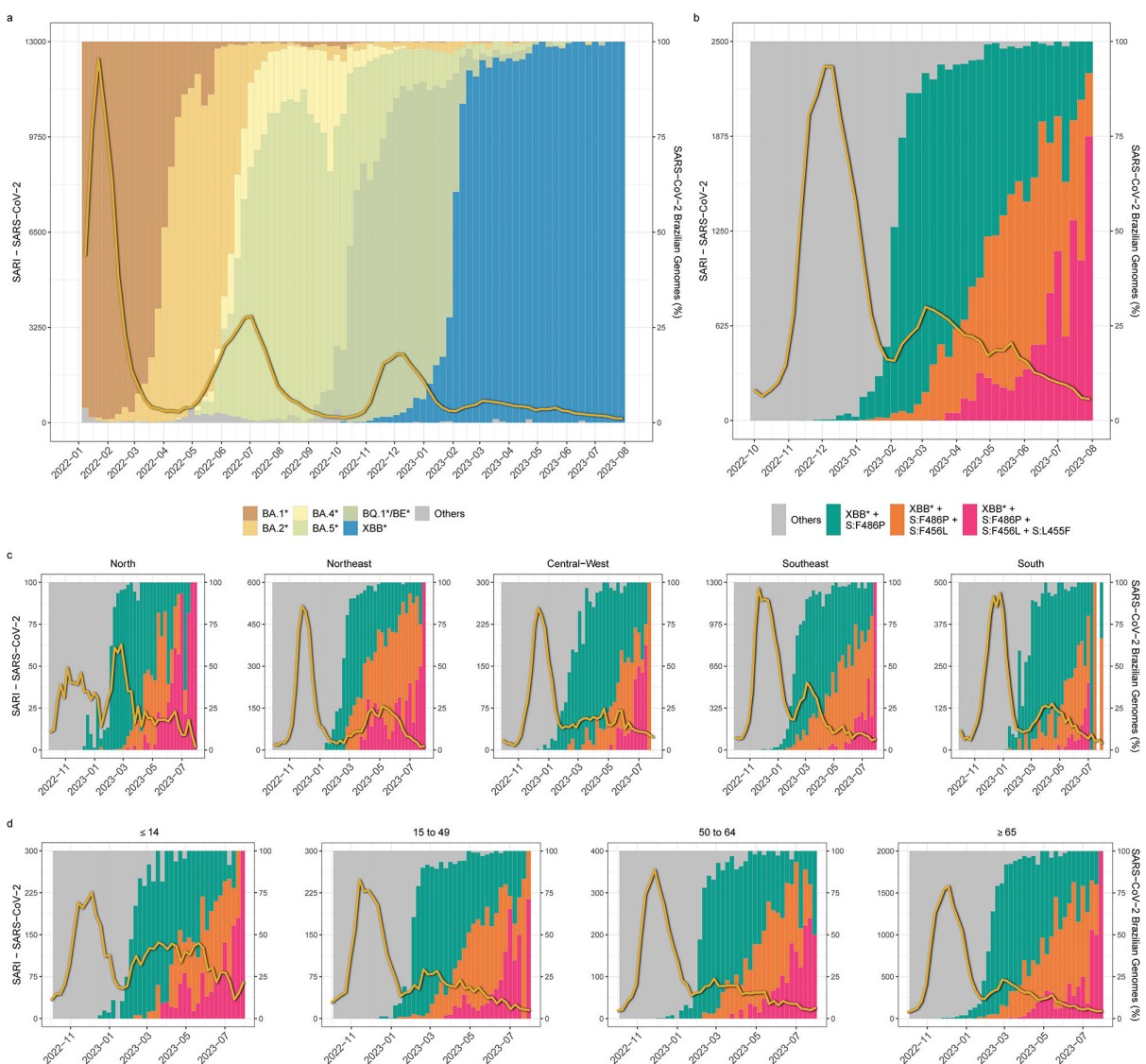

**FIG 3** Severe acute respiratory infection (SARI) detection in Brazil. (A) Evolution of SARI cases (yellow line) in Brazil between January 2022 and July 2023, plotted under the backdrop of SARS-CoV-2 Omicron lineage dissemination in the country, according to the color code at the panel's bottom. (B and C) Evolution of SARI cases in Brazil between October 2022 and July 2023, plotted under the backdrop of the distinct XBB* groups emergence and dissemination in Brazil (B), its five composing regions (C), and distinct age groups (D). The non-XBB and XBB* lineages lacking the S:F486P, S:F456L, and S:L455F mutations are aggregated and classified under the category "others." In all graphs, genomic and epidemiological data are plotted by epidemiological week.

XBB* + F486P + F456L + L455L lineages (Fig. 3B). The number of SARI-COVID cases at the peak of the XBB* wave was approximately 90%, 80%, and 70% lower than that observed during the peak of BA.1*, BA.5*, and BQ.1* waves, respectively. Moreover, the number of SARI-COVID notifications was reduced by 60% from January–June to July–December 2022 and by another 60% from July–December 2022 to January–July 2023.

When examining the incidence of SARI-COVID hospitalizations in the context of the spread of XBB* lineages across different Brazilian regions and age groups, a generally analogous trend was discerned, albeit with some variations (Fig. 3C and D). In the Northeastern, Central-Western, and Southern regions, the surge of SARI-COVID cases was temporally shifted to May. Moreover, the peak of the XBB* wave in the Northeastern region coincided with the dominance of the XBB* + F486P + F456L lineages, unlike the other regions that follow the national pattern. While the BQ.1* lineages dominated the Omicron wave in late 2022 in most Brazilian regions, SARS-CoV-2 cases in the Northern region were driven by a diverse mixture of BA.5-descendent lineages (BE.9, BQ.1*, and DL.1), and the peaks of both the BE.9/BQ.1*/DL.1 and XBB* waves were roughly similar and considerably lower than the peaks observed during the previous BA.1* and BA.5* waves. Finally, individuals under 15 years old exhibit a higher number of SARI cases during the XBB* dissemination period ($n = 2,655$) compared to what was observed during the BQ.1* dissemination period ($n = 2,090$), in contrast to all other age groups that displayed a significant decrease in SARI cases (50%–60%). Additionally, the number of SARI cases during the XBB* wave (February to July 2023) in individuals under 15 years old ($n = 2,655$, 23%) exceeded that of the age groups 15–49 ($n = 1,276$, 11%) and 50–64 ($n = 1,420$, 12%) and only fell below the count in individuals over 64 years old ($n = 6,348$, 54%). These data support that the dissemination of XBB* lineages in Brazil did not bring up a significant worsening of the SARS-CoV-2 epidemiological scenario in the country. It is noteworthy, however, that the incidence of SARI cases during the XBB* wave was higher among children (under 15) compared to young or middle-aged adults (aged 15–64), in contrast to previous Omicron waves.

## DISCUSSION

Our study reveals that the SARS-CoV-2 XBB epidemic in Brazil in 2023 was driven by multiple introductions followed by dissemination and evolution of different local lineages. The local evolution of XBB lineages in Brazil was characterized by the stepwise accumulation of mutations in the RBD of the Spike protein, which increase the immune escape and affinity to ACE2. It showed remarkable convergence with the molecular evolution pathway followed by other Omicron XBB lineages spreading in different countries worldwide (1, 6).

After being first detected in Brazil in October 2022, XBB* represented a small fraction (<5%) of the country's SARS-CoV-2 circulating lineages until early 2023, and the first XBB lineages to become prevalent in Brazil were those harboring the mutation S:F486P. The major XBB* + F486P lineages that arose and spread within Brazil were the XBB.1.5.86, XBB.1.5.102, XBB.1.18.1, and XBB.1.4.2. These Brazilian XBB* + F486P lineages probably started to be disseminated in Brazil between mid-October and late December 2022. They became the prevalent SARS-CoV-2 variants between January and February 2023, coinciding with the takeover of XBB* variants in Brazil. On a global scale, the XBB* + F486P genomes became the majority (~50%) among SARS-CoV-2 genomes submitted to GISAID (https://gisaid.org/) by February 2023, reached a peak of prevalence in May (~85%), and gradually decreased after that. These findings corroborate that XBB lineages were able to substitute the previously dominant BQ.1*/BE* lineages circulating in Brazil and other Western countries only after acquiring the S:F486P mutation that significantly increased the ACE2 binding affinity concerning ancestral XBB variants (21, 23–27). Our findings support that the average $R_{RI}$ of the XBB* + F486P lineages in Brazil was about 1.24 times higher than that of other cocirculating lineages, such as BQ.1* and ancestral XBB lineages, which is in line with the described average growth advantage of the

XBB.1.5 lineage w.r.t. XBB.1 and BQ.1* lineages circulating in the United States (1.23–1.26 times higher) (24).

The following step in the local evolution of XBB variants in Brazil was the acquisition of mutation S:F456L. A new Brazilian lineage carrying the mutation S:F456L, designated as FE.1*, arose in December 2022 from the XBB* + F486P lineage XBB.1.18.1. This lineage was first detected in Brazil in late January 2023 and gradually replaced XBB* + F486P lineages, becoming the most prevalent XBB variant by May 2023. The sequential accumulation of Spike mutations F486P and F456L in the Brazilian lineages XBB.1.18.1 and FE.1* reproduced the pattern observed in other XBB lineages that spread worldwide, such as the lineages XBB.1.9.2 and EG.5*. On a global scale, XBB* genomes carrying mutations S:F486P + F456L were first collected in November 2022 but displayed limited frequency (<5%) until May 2023. Subsequently, a fast surge in their prevalence propelled them to near-majority status (40%) by August 2023. The most prevalent XBB* + F486P + F456L lineage globally was the EG.5* (~55%), followed by FL* (~10%), FE.1* (~10%), and multiple ($n$ = 90) others XBB* lineages. Mutation S:F456L increases immune escape while reducing the affinity to ACE2 (17, 31, 33–35). Our findings support that the average $R_{RI}$ of the XBB* + F486P + F456L lineages circulating in Brazil was 1.07 times higher than that of XBB* + F486P variants. Similar values of growth advantage were described for the FE.1* (1.07–1.12 times higher) and EG.5.1 (1.15–1.23 times higher) lineages concerning the XBB.1.5 lineage in different countries from Asia (China, Japan, Singapore, and South Korea) and North America (Canada and USA) (31).

Between mid-February and early March 2023, XBB.1.5-descendant lineages carrying the Spike mutations F456L plus L455F arose in Brazil. This group's most prevalent Brazilian lineage was the XBB.1.5.70, along with its GK.1* descendants' lineages, followed by the lineages JD.1*. These lineages were detected in Brazil in March 2023 and surpassed other XBB* variants by June to July 2023. While mutations F456L and L455 individually reduce Spike's affinity for the ACE2 receptor, their simultaneous presence within the XBB.1.5 backbone collectively confers both increased ACE2 binding and immune escape properties (41, 64). Our analysis found that the $R_{RI}$ of the XBB* + F486P + F456L + L455F lineages in Brazil was 1.19 and 1.11 times higher than that of XBB* + F486P and XBB* + F486P + F456L lineages, respectively, providing the first estimates of growth advantage for these mutants. As of August 2023, XBB* genomes carrying mutations S:F486P + F456L + L455F still displayed a low global prevalence (~5%), and most genomes were assigned to the XBB.1.5.70 + GK* lineages in South America, North America, Europe, and Africa (60-80%) and to the HK.3 (EG.5.1.1.3) lineage in Asia (~50%). However, considering the ability of XBB* + F486P + F456L + L455F lineages to outcompete other XBB* lineages in Brazil and the convergent patterns of XBB evolution in different settings, it is reasonable to anticipate both the global increase in prevalence of these triply mutant lineages in the upcoming weeks and the emergence of mutations F456L + L455F in multiple XBB + F486P backbones.

The pronounced antigenic distance of XBB* lineages to the ancestral B.1 and other Omicron subvariants significantly reduced the efficacy of previous immunity to prevent XBB* infections (65–67). Thus, dissemination of XBB* lineages drove new epidemic waves in several countries (4, 30, 38, 39, 68). In Brazil, the growth advantage of XBB lineages carrying Spike mutations F486L, F486L + F456L, and F486L + F456L + L455F was sufficient to outcompete the BQ.1*/BE* lineages. However, their dissemination has had little epidemiological impact in terms of severe cases. SARI-COVID notifications displayed a modest increase during the spread of XBB* lineages in Brazil, much lower than during previous BA.1*, BA.5*, and BQ.1*/BE* waves. This scenario was not the result of a less intrinsic virulence because infection with XBB* subvariants results in similar disease severity to infection with other Omicron subvariants (4, 30, 69). The progressive reduction of severe cases across successive Omicron waves in Brazil since early 2022 indicates that population hybrid immunity has been increasingly efficient in preventing new upsurges of severe cases despite the continuous emergence of highly immune-evasive viral variants (65, 70). Repeated Omicron infections may also broaden the breadth

of antibody response and increase the neutralizing titers against highly immune-evasive variants (34), which is consistent with the observation that reduction in SARI cases from BQ.1*/BE* to XBB* waves in Brazil was higher in adult individuals than in the youngest (<15 years old) ones. It is also possible that reinfections due to waning immunity were limited in the XBB* wave by the short time interval of ~3 months from the previous BQ.1*/BE* wave, which was half the interval observed between Omicron waves in Brazil in 2022.

Although the diagnostic of SARS-CoV-2 cases in Brazil and most countries is no longer as exhaustive as it was during the first years of the pandemic, our findings reinforce the concept that the epidemiologic impact of SARS-CoV-2 lineage turnovers may greatly vary across different settings according to the previous immune experience of local populations. Countries such as Singapore and Malaysia did not experience a BQ.1*/BE* wave, and the magnitude of the XBB wave in late 2022 was similar to the previous BA.4/BA.5 wave occurring around 6 months earlier (4, 68). By contrast, Western countries that displayed BA.5* followed by BQ.1* waves during 2022 experienced transient surges in cases, hospital admissions, and deaths due to XBB* infections along 2023, particularly following the emergence of the EG.5* lineage, but the overall absolute numbers remained lower than during previous Omicron waves (30, 39, 39, 71, 72). These modest epidemiological consequences stemming from the substitution of BQ.1*/BE* lineages by XBB* in Brazil bears a resemblance to the replacement of the Gamma variant by the Delta variant during mid-2021 that also occurred without a significant upsurge of SARS-CoV-2 cases and deaths (73). Of note, the estimated average $R_{RI}$ of Delta w.r.t. Gamma (1.32) (73) was similar to the average $R_{RI}$ of XBB* concerning other co-circulating lineages (1.24–1.48) here estimated. Thus, the epidemiological impact of the spread of a new SARS-CoV-2 variant in a given region is influenced by an intricate interplay of factors that extend beyond the virus's transmissibility alone.

Predicting the antigenic evolution of XBB lineages in Brazil is a difficult task. By the time of writing this article, however, a new GK.1* and JD.1* sub-lineages harboring additional RBD mutations were identified in Brazil since early July. The GK.1.4 lineage harbors the RBD mutation K478R, while the GK.1.2.1, GK.1.8, and JD.1.1 harbor the convergent RBD mutation A475V. Interestingly, compared with XBB.1.5, the mutant XBB.1.5 + K478R exhibited a 1.7-fold higher infectivity in CaLu-3 cells and similar antibody evasion properties (17, 41). Meanwhile, the mutant XBB.1.5 + L455F + F456L + A475V could evade almost all of class 1 antibodies and is more immune evasive than XBB.1.5 + L455F + F456L but displays lower ACE2 binding (74). Some fast-growing XBB lineages displayed the RBD mutation K478R in combination with F486 (such as XBB.1.16), F486P + F456L (such as XBB.1.16.6 and FL.1.5.1), or F486P + F456L + L455F (such as JF.1). Similarly, multiple XBB lineages other than GK.1* and JD.1.1 also harbor the RBD mutation A475V in combination with mutations F486P + F456L + L455F (such as FL.15.1.1, GK.3.1, GK.4, GK.8.1, GW.5.1.1, and GW.5.3.1). Thus, the XBB lineages with the quadruple RBD mutant profiles L455F + F456L + K478R + F486P and L455F + F456L + A475V + F486P might display some growth advantage compared to double or triple RBD mutant XBB lineages currently prevalent, and their dissemination in Brazil and elsewhere should be closely monitored.

The major limitation of this study revolves around the availability of XXB* samples over the 10 months analyzed in various Brazilian states. Temporally, the majority of XXB* genomes were collected from February to May, constituting approximately 80% of the data set ($n$ = 5,611). As we progressed into June and July, the availability of sequences dropped significantly, accounting for only about 10% ($n$ = 797) of the total XBB* data set, and only 11 out of the 27 Brazilian states were sampled during July. This time frame corresponded with the period of expansion of the XBB* + F486P + F456L + LF455F group, and the prevalence of those triply mutated XBB* variants in Brazil could thus be underestimated. From a spatial perspective, the states of Amazonas ($n$ = 1,634), São Paulo ($n$ = 991), Bahia ($n$ = 591), and Santa Catarina ($n$ = 491) contributed with the most viral genomes, surpassing the combined contribution of the remaining 23 states.

This imbalanced geographic sampling hindered our ability to perform phylogeographic inferences at the sub-national level. This fact underscores the pressing need for an ongoing and comprehensive SARS-CoV-2 genomic surveillance system in Brazil that provides timely and spatially representative data on the SARS-CoV-2 genomic composition in various regions. It is important to note that as of mid-2023, these crucial data remain unavailable in several states.

In summary, our analysis shows the ongoing emergence of new XBB* lineages in Brazil during 2023, characterized by the gradual accumulation of mutations within the RBD region of the Spike protein and the steadily growing competitive edge. Mutations S:F486P + F456L + LF455F combined resulted in the XBB* variants with the highest transmissibility detected in Brazil in the first half of 2023. Notably, Brazil is the first country to have a high prevalence of XBB* + F486P + F456L + L455F lineages, and real-world transmissibility data obtained here support the potential of this new Omicron subvariant to spread globally. Notably, despite the significant growth advantage of XBB* lineages compared to other Omicron subvariants currently circulating, their impact on the epidemiological landscape of severe SARS-CoV-2 cases in Brazil has been less pronounced than that of the previously prevalent Omicron lineages. These findings underscore the importance of sustained molecular surveillance of SARS-CoV-2 cases to ensure rapid detection of new emergent viral variants with high transmissibility, pathogenicity, or both in Brazil and also revealed the importance of considering both virologic characteristics and population immune landscapes to predict the local impact of newly emergent SARS-CoV-2 lineages.

## ACKNOWLEDGMENTS

We gratefully acknowledge all data contributors, i.e., the originating laboratories responsible for obtaining the specimens and the submitting laboratories for generating the genetic sequence and metadata and sharing via the GISAID Initiative, on which this research is based. The authors also wish to thank all the healthcare workers and scientists who have worked hard to deal with this pandemic threat. In addition, we appreciate the support of the Respiratory Viruses Genomic Surveillance Network of the General Laboratory Coordination (CGLab) of the Brazilian Ministry of Health (MoH) and Brazilian Central Laboratory States (LACENs).

For members of the FIOCRUZ COVID-19 Genomics Surveillance Network, see the supplemental material.

This study was supported by the Department of Science and Technology (DECIT) of the Brazilian Ministry of Health (MoH); CGLab/MoH (General Laboratories Coordination of Brazilian Ministry of Health); UK Health Security Agency (UKHSA) by the New Variant Assessment Platform (NVAP) project; the Japan International Cooperation Agency (JICA); CVSLR/FIOCRUZ (Coordination of Health Surveillance and Reference Laboratories of Oswaldo Cruz Foundation); Centers for Disease Control and Prevention (CDC) grant; CNPq COVID-19 (MCTI402457/2020-0 and 403276/2020–9); INOVA Fiocruz (VPPCB-005-FIO-20-2 and VPPCB-007-FIO-18-2-30); FAPERJ (E26/210.196/2020); FAPEAM (Rede Genômica de Vigilância em Saúde-REGESAM); FAPEAM (INICIATIVA AMAZÔNIA + 10 [grant: 01.02.016301.00439/2023-70]); FAPEAM/INOVA FIOCRUZ INOVAÇÃO NA AMAZÔNIA (Chamada Pública no. 04/2022); NPI EXPAND–U.S. Agency for International Development (USAID) implemented by Palladium (7200AA19CA00015), Centers for Disease Control and Prevention (CDC Grant Award 002174), and CNPQ CABBIO (grant number 423857/2021-5); and FAPERJ (grant number E-26/211.125/202). P.C.R. had support from a CNPQ productivity research fellowship (311759/2022-0). M.M.S. had support from a CNPq productivity research fellowship (313403/2018-0). F.G.N. had support from a CNPq productivity research fellowship (306146/2017-7). G.B. had support from FAPERJ (grant number E-26/202.896/2018) and CNPq productivity research fellowship (304883/2020-4). I.A. had support from FAPERJ-Fundação Carlos Chagas Filho de Amparo à Pesquisa do Estado do Rio de Janeiro (grant SEI-260003/019669/2022). This research was supported by FINDINGS Project (Project for the Enhancement of

Genomic Monitoring Network for Covid-19) agreed upon between FIOCRUZ (Oswaldo Cruz Foundation), Brazilian Cooperation Agency and JICA (Japan International Cooperation Agency) on 13 March 2023. This study was partially supported by the Coordenação de Aperfeiçoamento de Pessoal de Nível Superior–CAPES-Finance Code 001.

The study was conceived and designed by G.B., F.G.N., and I.A. F.M., R.K., T.G., P.C.R., and F.G.N. contributed to diagnostics and sequencing analysis. K.I. contributed to the instantaneous reproduction number estimations. M.G., F.C.D.C., and W.A.F.D.A. worked on the retrieval and analysis of Brazilian epidemiological data. F.G.N. and M.M.S. contributed to laboratory management and obtaining financial support. The bioinformatics analysis was performed by I.A. and S.S. I.A. and G.B. wrote the first draft, and all authors contributed and approved the final manuscript.

## AUTHOR AFFILIATIONS

[1]Laboratório de Arbovírus e Vírus Hemorrágicos, Instituto Oswaldo Cruz, Fiocruz, Rio de Janeiro, Brazil

[2]Grupo de Métodos Analíticos em Vigilância Epidemiológica, Fiocruz, Rio de Janeiro, Brazil

[3]International Institute for Zoonosis Control, Hokkaido University, Hokkaido, Japan

[4]Laboratório de Virologia Molecular, Instituto Carlos Chagas, Fiocruz, Curitiba, Brazil

[5]Fiocruz, Fortaleza, Brazil

[6]Instituto Gonçalo Moniz, Fiocruz, Salvador, Brazil

[7]Departamento do Programa Nacional de Imunizações, Coordenação-Geral de Vigilância das doenças imunopreveníveis, Secretaria de Vigilância em saúde e ambiente, Brasília, Brazil

[8]Laboratório de Vírus Respiratórios, Exantemáticos, Enterovírus e Emergências Virais, Instituto Oswaldo Cruz, Fiocruz, Rio de Janeiro, Brazil

[9]Núcleo de Vigilância de Vírus Emergentes, Reemergentes ou Negligenciados, Laboratório de Ecologia de Doenças Transmissíveis na Amazônia, Instituto Leônidas e Maria Deane, Fiocruz, Manaus, Brazil

## AUTHOR ORCIDs

Ighor Arantes http://orcid.org/0000-0002-4131-5338
Fabio Miyajima http://orcid.org/0000-0002-1347-4825
Gonzalo Bello http://orcid.org/0000-0002-2724-2793

## FUNDING

| Funder | Grant(s) | Author(s) |
| --- | --- | --- |
| Fundação Carlos Chagas Filho de Amparo à Pesquisa do Estado do Rio de Janeiro (FAPERJ) | E-26/202.896/2018 | Gonzalo Bello |
| Conselho Nacional de Desenvolvimento Científico e Tecnológico (CNPq) | 304883/2020-4 | Gonzalo Bello |
| Fundação Carlos Chagas Filho de Amparo à Pesquisa do Estado do Rio de Janeiro (FAPERJ) | SEI-260003/019669/2022 | Ighor Arantes |
| Conselho Nacional de Desenvolvimento Científico e Tecnológico (CNPq) | 306146/2017-7 | Felipe Gomes Naveca |
| Conselho Nacional de Desenvolvimento Científico e Tecnológico (CNPq) | 313403/2018-0 | Marilda Mendonça Siqueira |

| Funder | Grant(s) | Author(s) |
|---|---|---|
| Conselho Nacional de Desenvolvimento Científico e Tecnológico (CNPq) | 311759/2022-0 | Paola Cristina Resende |
| Conselho Nacional de Desenvolvimento Científico e Tecnológico (CNPq) | MCTI402457/2020-0, 403276/2020-9, 423857/2021-5 | COVID-19 Fiocruz Genomic Surveillance Network Genomahcov |
| Fundação Oswaldo Cruz (FIOCRUZ) | VPPCB-005-FIO-20-2, VPPCB-007-FIO-18-2-30 | COVID-19 Fiocruz Genomic Surveillance Network Genomahcov |
| Fundação Carlos Chagas Filho de Amparo à Pesquisa do Estado do Rio de Janeiro (FAPERJ) | E26/210.196/2020, E-26/211.125/202 | COVID-19 Fiocruz Genomic Surveillance Network Genomahcov |
| Fundação de Amparo à Pesquisa do Estado do Amazonas (FAPEAM) | 01.02.016301.00439/2023-70 | COVID-19 Fiocruz Genomic Surveillance Network Genomahcov |
| United States Agency for International Development (USAID) | 7200AA19CA00015 | COVID-19 Fiocruz Genomic Surveillance Network Genomahcov |
| HHS | Centers for Disease Control and Prevention (CDC) | 002174 | COVID-19 Fiocruz Genomic Surveillance Network Genomahcov |

## AUTHOR CONTRIBUTIONS

Ighor Arantes, Conceptualization, Data curation, Formal analysis, Investigation, Methodology, Visualization, Writing – original draft, Writing – review and editing | Marcelo Gomes, Data curation, Methodology, Writing – review and editing | Kimihito Ito, Data curation, Formal analysis, Methodology, Writing – review and editing | Sharbilla Sarafim, Formal analysis, Writing – review and editing | Tiago Gräf, Funding acquisition, Writing – review and editing | Fabio Miyajima, Funding acquisition, Writing – review and editing | Ricardo Khouri, Funding acquisition, Writing – review and editing | Felipe Cotrim de Carvalho, Data curation, Writing – review and editing | Walquiria Aparecida Ferreira de Almeida, Data curation, Writing – review and editing | Marilda Mendonça Siqueira, Funding acquisition, Project administration, Writing – review and editing | Paola Cristina Resende, Funding acquisition, Writing – review and editing | Felipe Gomes Naveca, Conceptualization, Funding acquisition, Writing – review and editing | Gonzalo Bello, Conceptualization, Data curation, Formal analysis, Funding acquisition, Investigation, Methodology, Project administration, Supervision, Writing – original draft, Writing – review and editing.

## DATA AVAILABILITY

The findings of this study are based on the analysis of 107,881 SARS-CoV-2 Brazilian genomes collected after 1 January 2022 and submitted to the GISAID database until 31 July 2023, accessible at https://doi.org/10.55876/gis8.231024oa. In our phylogeographic analysis, we additionally downloaded 3,132 global references from the same period available at https://doi.org/10.55876/gis8.231024qe.

## ADDITIONAL FILES

The following material is available online.

### Supplemental Material

**Supplemental Material (Spectrum03831-23-S0001.pdf).** Supplemental figures, tables, and information.

### Open Peer Review

**PEER REVIEW HISTORY (review-history.pdf).** An accounting of the reviewer comments and feedback.

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
