## [Reviewer comments · Microbiology Spectrum]

Microbiology Spectrum

Spatiotemporal dynamics and epidemiological impact of SARS-CoV-2 XBB lineages dissemination in Brazil in 2023

Ighor Arantes, Marcelo Gomes, Kimihito Ito, Sharbilla Sarafim, Tiago Gräf, Fabio Miyajima, Ricardo Khouri, Felipe Carvalho, Walquiria de Almeida, Marilda Siqueira, Paola Resende, Felipe Naveca, Gonzalo Bello, and COVID-19 Fiocruz Genomic Surveillance Network Genomahcov

Corresponding Author(s): Ighor Arantes, Fundacao Oswaldo Cruz

Review Timeline:

Submission Date:	November 6, 2023
Editorial Decision:	December 1, 2023
Revision Received:	December 21, 2023
Accepted:	December 27, 2023

Editor: Juan Ludert

Reviewer(s): Disclosure of reviewer identity is with reference to reviewer comments included in decision letter(s). The following individuals involved in review of your submission have agreed to reveal their identity: Nabil Benazi (Reviewer #1)

Transaction Report:

DOI: <https://doi.org/10.1128/spectrum.03831-23>

Re: Spectrum03831-23 (Spatiotemporal dynamics and epidemiological impact of SARS-CoV-2 XBB lineages dissemination in Brazil in 2023)

Dear Dr. Ighor Arantes:

Thanks for submitting your work to Microbiology Spectrum. Your manuscript was reviewed by two experts in the field, who agree that the manuscript is of interest and merits publication after modifications. You will find their comments below, as well as instructions from the Spectrum editorial office.

Revision Guidelines

Sincerely,
Juan Ludert
Editor
Microbiology Spectrum

Reviewer #1 (Comments for the Author):

The paper in its form and content was very well presented, however I have some questions which should be explained by the kind authors.

In the "Estimates of relative instantaneous reproduction number (R_{RI})" section:

Question 1: why did you assume that the rate from $R_{Ai}(t)$ to $R_I(t)$ is constant over time for each lineage A_i ?

Question 2: why did you set the parameters of the gamma distribution as $\alpha = 4.03$ and $\theta = 0.737$ in your distribution model. (How did you get these values)?

Reviewer #2 (Comments for the Author):

This study explore the Spatiotemporal distribution of the three mutations S:F486P, S:F456L, and S:L455F in Brazil, as well as the analysis of relative instantaneous reproduction number estimations and SARI. I just have a few questions that may need the authors to be clarified:

1. For the non-Brazilian XBB genome sequences and dataset composition, are all those sequences only included the S:F486P, S:F456L, and S:L455F? Or they not only included the above three mutations but with other mutations?
2. There is still not that clear why the four criteria could identify the sari-covid cases in the SIVEP-Gripe database.
3. I would suggest to increase the resolution and front size in all the figures. It is difficult to read the contents in the figures with low resolution and small front size.
4. There are time-scaled trees for each lineage in the supplementary figures. You mentioned in the method that you also used the IQTREE for the ML phylogenetic analysis, however, I did not see these results in the manuscripts. The reason I mentioned this is that I would like to see the tree of the overall database.
5. Though the three mutations are important and is the main focus in this study, I would still suggest that the authors could provide some briefly discussion of other mutations that could infect the XBB evolution and dissemination in Brazil.

Reviewer #1 (Comments for the Author):

The paper in its form and content was very well presented, however I have some questions which should be explained by the kind authors.

In the "Estimates of relative instantaneous reproduction number (R_{RI})" section:

Question 1: why did you assume that the rate from R_{Ai}(t) to R_I(t) is constant over time for each lineage A_i?

A previous study by Piantham and Ito has shown that observation data on the SARS-COV-2 variant replacement were well-modeled by assuming that the rate of the effective reproduction number of a variant to that of another variant was constant over time. We added this description in the revised manuscript. (Lines 203-206) "A previous study (56) has shown that observation data on the SARS-COV-2 variant replacement were well-modeled by assuming that the rate of the effective reproduction number of a variant to that of another variant was constant over time."

Question 2: why did you set the parameters of the gamma distribution as $\alpha = 4.03$ and $\theta = 0.737$ in your distribution model. (How did you get these values)?

The parameters of the gamma distribution were determined so that the distribution has the same mean and variance to the lognormal distribution estimated by Park et al. This was described in the paragraph above the formula of $g(i)$. To clarify this more, we added the following description to the revised manuscript. (Lines 223-224) "Parameters of the gamma distribution were set to be $\alpha=4.03$ and $\theta=0.737$, so that the gamma distribution has the same mean and variance to the lognormal distribution estimated by Park et al. (60)."

Reviewer #2 (Comments for the Author):

This study explore the Spatiotemporal distribution of the three mutations S:F486P, S:F456L, and S:L455F in Brazil, as well as the analysis of relative instantaneous reproduction number estimations and SARI. I just have a few questions that may need the authors to be clarified:

1. For the non-Brazilian XBB genome sequences and dataset composition, are all those sequences only included the S:F486P, S:F456L, and S:L455F? Or they not only included the above three mutations but with other mutations?

In each of the major Brazilian XBB lineages that carried Spike mutations F486P, F456L, and L455F among their synapomorphies, we composed a larger dataset of non-Brazilian sequences carrying those same mutations in order to understand the lineage's space-time dynamics. This larger dataset included all XBB sequences available worldwide from a given lineage that displayed the selected synapomorphic mutations as well as a small fraction of unidentified positions ($n < 5\%$). These did not preclude the included genomes from carrying other Spike mutations (different from F486P, F456L, and L455F).*

2. There is still not that clear why the four criteria could identify the sari-covid cases in the SIVEP-Gripe database.

The criteria used to identify SARI-COVID cases were based on the case definition used for SARI in Brazil, according to the Health Surveillance Guide, plus the requisite of a positive RT-PCR test for SARS-CoV-2. The original manuscript was modified to make this question clearer. In line 252, we described the parameters for the definition of a SARI case and later in this paragraph (Lines 258-259) is presented the requisite of a positive RT-PCR test to classify the case as SARI-COVID.

3. I would suggest to increase the resolution and font size in all the figures. It is difficult to read the contents in the figures with low resolution and small front size.

We apologize for the low quality of the figures submitted with the original manuscript. In the current version, the resolution was increased in all figures and we also increased the font in the figures. All figures were also submitted as individual files, and their legends are present at the end of the manuscript.

4. There are time-scaled trees for each lineage in the supplementary figures. You mentioned in the method that you also used the IQTREE for the ML phylogenetic analysis, however, I did not see these results in the manuscripts. The reason I mentioned this is that I would like to see the tree of the overall database.

*In the process of inference of the time-scales trees, ML trees were used as fixed topologies, but these were not included in the final version of the manuscript. In the original supplementary material, the time-scaled trees were included as **Supplementary Figures 1-4**, however, to comply with the reviewer's suggestion, these are now **Supplementary Figures 4-7**, and the ML trees were included as **Supplementary Figures 2-3**.*

5. Though the three mutations are important and are the main focus in this study, I would still suggest that the authors could provide some brief discussion of other mutations that could infect the XBB evolution and dissemination in Brazil.

*The S:F486P, F456L and L455F mutations were analyzed since they combined (i) previous descriptions of their effect on the increase of SARS-CoV-2 transmissibility and (ii) a pronounced expansion (>25%) among the Brazilian genomic profile by means of independent emergences in multiple lineages. This last observation may not have been clear in the original manuscript, so we included a mention to this in the Results section (Lines 274-276) and presented it visually in an additional supplementary figure (**Supplementary Figure 1**) in which we measured the frequency of all Spike mutations detected in Brazilian XBB* genome sequences, excluding those mutations already synapomorphic to lineage XBB.1. Moreover, in the Discussion section of the manuscript (Lines 501-519) we discussed the potential relevance of mutations S:K478R and S:A475V, since they also had previous descriptions of transmissibility increase, although they lacked, at the the time of the manuscript submission, an increase in prevalence in the country's genomic profile.*

Re: Spectrum03831-23R1 (Spatiotemporal dynamics and epidemiological impact of SARS-CoV-2 XBB lineages dissemination in Brazil in 2023)

Dear Dr. Ighor Arantes:

Congratulation! Your manuscript has been accepted, and I am forwarding it to the ASM production staff for publication. Your paper will first be checked to make sure all elements meet the technical requirements. ASM staff will contact you if anything needs to be revised before copyediting and production can begin. Otherwise, you will be notified when your proofs are ready to be viewed.

Sincerely,
Juan Ludert
Editor
Microbiology Spectrum